Chromatin accessibility landscape of stromal subpopulations reveals distinct metabolic and inflammatory features of porcine subcutaneous and visceral adipose tissue

Sun Wenyang 1 2
Zhang Tinghuan 2
Hu Silu 3
Tang Qianzi 3
Long Xi 2
Yang Xu 4
Gun Shuangbao gunsbao056@126.com 1
Chen Lei sicau.chen@gmail.com 2
1 College of Animal Science and Technology, Gansu Agricultural University , Lanzhou , Gansu , China
2 Key Laboratory of Pig Industry Sciences (Ministry of Agriculture), Chongqing Academy of Animal Science , Chongqing , China
3 Institute of Animal Genetics and Breeding, College of Animal Science and Technology, Sichuan Agricultural University , Chengdu , Sichuan , China
4 College of Nursing, Ningxia Medical University , Yinchuan , Ningxia , China
Uversky Vladimir
Electronic publication date: 2022 May 24
Publication date: 2022
Volume: 10
Electronic Location ID: e13250
Received 2021 Nov 12; Accepted 2022 Mar 21
Copyright: ©2022 Sun et al.
Copyright year: 2022
Copyright holder: Sun et al.
License: This is an open access article distributed under the terms of the Creative Commons Attribution License, which permits unrestricted use, distribution, reproduction and adaptation in any medium and for any purpose provided that it is properly attributed. For attribution, the original author(s), title, publication source (PeerJ) and either DOI or URL of the article must be cited.
License URL: https://creativecommons.org/licenses/by/4.0/

Keywords: Subcutaneous adipose tissues, Visceral adipose tissues, Stromal vascular fraction, Chromatin accessibility, Pig

Funding: National Natural Science Foundation of China 31771376 Chongqing Scientific Research Institution Performance Incentive Project 19536 19540 20520 Chongqing Fundamental Research Funds cstc2017jxjl0057 2015cstc-jbky-00101 This work was supported by the National Natural Science Foundation of China (No. 31771376), the Chongqing Scientific Research Institution Performance Incentive Project (No. 19536, No. 19540, No. 20520) and Chongqing Fundamental Research Funds (No. cstc2017jxjl0057, No. 2015cstc-jbky-00101). The funders had no role in study design, data collection and analysis, decision to publish, or preparation of the manuscript.

==============================
Background

Fat accumulation in visceral adipose tissue (VAT) confers increased risk for metabolic disorders of obesity, whereas accumulation of subcutaneous adipose tissue (SAT) is associated with lower risk and may be protective. Previous studies have shed light on the gene expression profile differences between SAT and VAT; however, the chromatin accessibility landscape differences and how the cis-regulatory elements govern gene expression changes between SAT and VAT are unknown.

Methods

Pig were used to characterize the differences in chromatin accessibility between the two adipose depots-derived stromal vascular fractions (SVFs) using DNase-sequencing (DNase-seq). Using integrated data from DNase-seq, H3K27ac ChIP-sequencing (ChIP-seq), and RNA-sequencing (RNA-seq), we investigated how the regulatory locus complexity regulated gene expression changes between SAT and VAT and the possible impact that these changes may have on the different biological functions of these two adipose depots.

Results

SVFs form SAT and VAT (S-SVF and V-SVF) have differential chromatin accessibility landscapes. The differential DNase I hypersensitive site (DHS)-associated genes, which indicate dynamic chromatin accessibility, were mainly involved in metabolic processes and inflammatory responses. Additionally, the Krüppel-like factor family of transcription factors were enriched in the differential DHSs. Furthermore, the chromatin accessibility data were highly associated with differential gene expression as indicated using H3K27ac ChIP-seq and RNA-seq data, supporting the validity of the differential gene expression determined using DNase-seq. Moreover, by combining epigenetic and transcriptomic data, we identified two candidate genes, NR1D1 and CRYM, could be crucial to regulate distinct metabolic and inflammatory characteristics between SAT and VAT. Together, these results uncovered differences in the transcription regulatory network and enriched the mechanistic understanding of the different biological functions between SAT and VAT.

Introduction

Obesity has become a global health problem (Gregg & Shaw, 2017). With the development of obesity, white adipose tissues (WATs) increasingly dysfunctional and release more free fatty acids and proinflammatory cytokines. Both elevated free fatty acids and proinflammatory cytokines in obesity play critical roles in the aetiology of obesity-related metabolic syndrome (Lee, Wu & Fried, 2013). It is widely appreciated that the obesity-related metabolic risks are most related to the pattern of fat distribution rather than the total amount of body fat (Despres & Lemieux, 2006; Lusis, Attie & Reue, 2008). Generally, two major anatomical depots of WATs are recognized: subcutaneous adipose tissue (SAT) and visceral adipose tissue (VAT). Increased accumulation of VAT is strongly associated with insulin resistance, type 2 diabetes mellitus, and metabolic syndrome (Karlsson et al., 2019; Marinou et al., 2014). By contrast, elevated amounts of SAT are associated with improvements in insulin sensitivity and a lower risk of developing type 2 diabetes (Misra et al., 1997; Snijder et al., 2003). Decreasing VAT mass can improves insulin sensitivity and glucose metabolism (Thorne et al., 2002), whereas reducing SAT mass does not (Klein et al., 2004). Indeed, SAT and VAT depots exhibit several differences, including the capacities of lipogenesis and lipolysis, gene expression, and adipokine secretion profiles (Karastergiou et al., 2013; Lafontan & Berlan, 2003). The metabolic dysfunctions seen in VAT are due to its higher lipolysis rate, which leads to an increase in the release of free fatty acids and the secretion of proinflammatory cytokines. Together, these two functions can lead to insulin resistance (Fontana et al., 2007; Wajchenberg, 2000). In contrast, the beneficial effects of SAT are related to its ability to take up more free fatty acids and convert them into storable fat via lipogenesis, which protects the organism from lipotoxicity (Frayn, 2002).

WAT is composed of two major cell groups: mature adipocytes and the stromal vascular fraction (SVF). The SVF contains adipocyte precursor cells, endothelial cells, hematopoietic-lineage cells, immune cells, stem cells, and stromal cells. The adipocyte precursor cells, with potential capacity of differentiation and adipogenesis, can differentiate into adipocytes. Thus, the SVF can modulate the lipid storage capacity of WAT by maintaining or increasing adipocyte numbers (Gray & Vidal-Puig, 2007). Moreover, the SVF is involved in inflammation of WAT by secreting inflammatory cytokines. The classically activated macrophages (M1 macrophages) from the SVF have been recognized as the main source of proinflammatory cytokines in WATs, such as tumor necrosis factor α (TNF-α), interleukin-6 (IL-6) and interleukin-1β (IL-1β), and as key participants in obesity-induced inflammation (Mathis, 2013). The heterogeneity between SAT and VAT could be attributed to differential behavior of cells in each WAT. Indeed, the SVF from SAT and VAT (S-SVF and V-SVF) also has an array of biological differences, including adipogenic ability (Tchkonia et al., 2006), immunomodulatory potentials (Silva & Baptista, 2019), metabolic characteristics (Lefevre et al., 2019), and gene expression profiles (Tchkonia et al., 2007). These differences in the SVF reinforce the different biological characteristics between SAT and VAT. It was suggested that the understanding the characteristics that lead to the depot-specific SVF could be pivotal in determining the roles of each depot in metabolic disorders (Peinado et al., 2010).

Differential gene expression profiles likely explain why SAT and VAT have different pathophysiological functions (Zhou et al., 2013), yet the roles of transcriptional regulation mechanisms in causing the depot-specific gene expression profiles and biological characteristics are not well understood. The precise coordination of the temporal and spatial regulation of gene expression is necessary to achieve tissue development (Agrawal, Heimbruch & Rao, 2018), and this regulation, in eukaryotes, is achieved through the interaction between the noncoding functional cis-regulatory elements in genomes and trans-acting factors. Characterizing the chromatin regulatory landscape is key to understanding the regulation of tissue-specific gene expression that governs tissue-specific biological processes (Agrawal, Heimbruch & Rao, 2018). Thus, revealing the differences in chromatin regulatory landscape between S-SVF and V-SVF could be crucial for understanding why SAT and VAT vary in their effects on metabolic disorders.

Mapping DNase I hypersensitive sites (DHSs) is a valuable method of chromatin accessibility and has been widely used to discover different types of cis-regulatory elements, including promoters, enhancers, silencers, insulators, and most transcription factor-binding sites (Gross & Garrard, 1988). DNase I treatment combined with high-throughput DNA sequencing (DNase-seq) has allowed comprehensive and accurate genome-wide identification of DHSs (Boyle et al., 2008). However, the conventional DNase-seq technique requires millions of cells, and, therefore, cannot be used on rare biological samples. Recently, however, a low-input DNase-sequencing (liDNase-seq) method has been developed that can be used on samples with few cells (Lu et al., 2016).

Recently, pigs (Sus scrofa) have been emerging as an exceptional biomedical model for studying human obesity (Rocha & Plastow, 2006; Spurlock & Gabler, 2008). Here, we established the chromatin accessibility landscape of the SVF from porcine SAT and VAT using liDNase-seq to identify depot-specific cis-regulatory elements and motifs in order to reveal the fundamental mechanisms regulating their differential gene expression profiles and biological characteristics. In addition, by integrating our DNase-seq with H3K27ac ChIP-sequencing (ChIP-seq) and RNA-sequencing (RNA-seq) datasets of porcine SAT and VAT, we revealed how the dynamic chromatin accessibility landscape works with H3K27ac modifications to regulate differential gene expression to modulate the different biological characteristics between SAT and VAT. These results can promote further research of obesity-related metabolic disease.

Materials and Methods

Ethics statement

All methods involving animal experiments were performed strictly in accordance with the recommendations for the Care and Use of Laboratory Animals of the National Institutes of Health of China. The protocol employed in this study was reviewed and approved by the Animal Ethical and Welfare Committee of Gansu Agricultural University (approval number: AEWC-GAU-2019-096).

Animals and sample collection

Two 150-day-old females Bama miniature pigs were chosen from the animal breeding facility of the Chongqing Academy of Animal Science in Chongqing, China. Pigs were placed on high-fat diet until body weight reached 165 ± 5 kg, and then the pigs were humanely sacrificed using electro-stunning followed by severance of blood vessels in the neck. All pigs used in this study were allowed access to feed and water ad libitum and lived under the same normal conditions. SVT (back fat) and VAT (retroperitoneal adipose) were excised from each carcass and used for subsequent experiments.

Isolation of adipose SVF

Three grams of WAT (SAT or VAT) from each pig was minced in a 50 mL tube and digested in 20 mL Hank’s balanced salt solution (Life Technologies) containing 1 mg/mL collagenase type I (Life Technologies) and 1.5% bovine serum albumin (Sigma-Aldrich) at 37 °C in a shaker at 200 rpm for 45 min. The digested homogenates were filtered through 100 µm cell strainer to remove undigested tissues and centrifuged at 800 × g for 15 min at 4 °C. The bottom SVF pellets were washed with 10 mL of PBS and passed through a 40 µm cell strainer to remove larger adipocytes and clumps, and recentrifuged at 800 × g for 15 min at 4 °C.

Low-input DNase-seq

We pooled the SVF (S-SVF or V-SVF) from two biological replicates (each one 5.0 × 104 cells). Approximately 1.0 × 105 SVF cells (S-SVF or V-SVF) were used for liDNase-Seq, performed as previously described (Lu et al., 2016). Briefly, SVF cells were digested in 36 µL lysis buffer (10 mM Tris-HCl, pH 7.5, 10 mM NaCl, 3 mM MgCl2, 0.1% Triton X-100) and incubated on ice for 5 min. DNase I (3.6 U; Roche) was added to final concentration of 0.1 U/µL and incubated at 37 °C for 5 min. Stop Buffer (80 µL; 10 mM TrisHCl, pH 7.5, 10 mM NaCl, 0.15% SDS, 10 mM EDTA) containing 2 µL Proteinase K (20 mg/mL, Life Technologies) and 20 ng circular carrier DNA (pUC19 DNA, Life technologies) was added and incubated at 50 °C for 1 h to stop the reaction. DNA was purified by phenol–chloroform (Sigma-Aldrich) extraction and precipitated with ethanol containing linear acrylamide (Life Technologies). The sequencing libraries were prepared using NEBNext Ultra DNA Library Prep Kit for Illumina (NEB). The libraries were amplified twice using PCR amplification. DNA fragments (160–300 bp) were purified using E-gel, and sequenced on an Illumina Novaseq 6000 platform with single-end 50 bp reads.

DNase-seq data analysis

DNase-seq reads were mapped to the pig genome (Sscrofa 11.1) using Bowtie2, allowing a single mismatch. Unmapped reads were iteratively aligned by trimming 5 bp each pass until reads were less than 26 bp. Redundant reads were removed if it mapped to the same location with the same orientation. The DHSs of each library were identified using MACS2, with a cutoff of P < 1 × 10−5. The differential DHSs were scored using DEGseq an adjusted P value of P < 0.01, and then assigned to genes using HOMER. Heat maps of the DNase-seq data were generated by deepTools2.

ChIP-seq and data analysis

The frozen SATs and VATs (with two biological replicates) were grounded into powder in liquid nitrogen. The powders were fixed in 1% formaldehyde (Sigma-Aldrich) at room temperature for 20 min, and the reaction was subsequently quenched by adding glycine to a final concentration of 0.2 M and incubating for 5 min. Nuclear lysates were sonicated using Covaris S220 (Adaptive Focused Acoustics®). Fixed chromatin was then incubated with an antibody against H3K27ac (Abcam) or M-280 sheep anti-rabbit IgG Dynabeads (Thermo Fisher) at 4 °C for 4 h. DNA was extracted using antibody–bead complexes, and ChIP-seq libraries were prepared according to standard Illumina protocols and further sequenced on an Illumina Novaseq 6000 platform with paired-end 150 bp reads. Reads were aligned to the Sscrofa 11.1 genome using Bowtie2 and filtered to remove duplicates with Picard and SAMTools. Peak calling was conducted using MACS2. Peaks were called if they passed a false discovery rate of 0.01 and were enriched over input. The reproducible peaks in two biological replicates were identified using irreproducible discovery rate (IDR) with a cutoff of P-value < 0.01. The differential analysis of peaks was performed using DESeq2 with a cutoff of P < 0.01. The differential peaks were assigned to genes using HOMER. Heat maps of the ChIP-seq reads were generated by deepTools2.

RNA-seq and data analysis

The total RNA of SATs and VATs (with two biological replicates) were isolated using the miRNeasy Mini Kit (Qiagen). Sequencing libraries were constructed using the TruSeq Stranded Total RNA Library Prep Kit (Illumina), and further sequenced on an Illumina Novaseq 6000 platform with paired-end 150 bp reads. Clean data were obtained by removing reads containing adapters, reads containing over 10% of poly-N (unrecognized bases), and low-quality reads (>50% of bases with Phred scores <5). Reads were mapped to the pig genome (Sscrofa 11.1) using Bowtie2. Gene expression levels were evaluated by the RPM (reads per million mapped reads) values of mRNA calculated using featureCounts. Differential gene expression analysis was performed using edgeR 3.31. The differentially expressed genes with a P-value of < 0.01 were identified. The R package RNASeqPower (http://bioconductor.org/packages/release/bioc/html/RNASeqPower.html) was used to estimate the power of differential expression analysis.

Gene ontology enrichment analysis and motif analysis

Gene Ontology (GO) enrichment analyses were performed with Metascape (http://metascape.org/gp/index.html#/main/step1). Enrichment of known motifs in differential DHSs was analyzed using Homer (http://homer.ucsd.edu/homer/motif/).

Results

Establishing chromatin accessibility landscape of S-SVF and V-SVF

To understand the characteristics of chromatin accessibility landscape in S-SVF and V-SVF, we isolated SVF from porcine SAT and VAT and used liDNase-seq method (pooling SVF form two biological replicates) to map DHSs in S-SVF and V-SVF. It is known that the RNA polymerase II complex binds double-stranded DNA and releases single-stranded DNA around the transcriptional start site (TSS) to establish a DNase I hypersensitive region. This region represents open chromatin, and therefore, would have abundant DHS signal intensity (Allen & Taatjes, 2015; Heinz et al., 2015). Thus, we examined the DHS signal intensity around the gene TSS within ± 2 kb. We found strong DHS signal intensity around the TSSs of numerous genes in both S-SVF and V-SVF (Fig. 1A). Using peak calling, 3930 and 2801 DHSs were identified in S-SVF and V-SVF, respectively (Table S1). To characterize these DHSs, we examined their genomic location on annotated genes (Fig. 1B and Table S2). The DHSs are highly enriched in proximal promoters (−2 kb to +100 bp from TSS) in S-SVF (50.9%). This ratio decreased in V-SVF (21.6%), and DHSs at gene bodies (exon and intron) and intergenic regions are increased in V-SVF, showing the different distribution pattern of DHSs between S-SVF and V-SVF. Furthermore, we found that CD34 and ALCAM (CD166), which are the molecular markers of SVF (Bora & Majumdar, 2017; Walmsley et al., 2015), have obvious DHS in distal gene promoters and TSS (Fig. 1C), indicating that our isolation of SVF and corresponding DNase-seq data are reliable.

Figure 1 Establishing chromatin accessibility landscapes in S-SVF and V-SVF.

(A) Heat map showing enrichment of normalized DNase-seq reads within ±2 kb of TSS in S-SVF and V-SVF. (B) The genomic distribution of the DHSs (the genome locations of DHSs form TSS). (C) Genome browser view of two SVF marker genes CD34 (in upper panel) and ALACM (in lower panel). Gray boxes indicate DHS sites.

Heterogeneity in chromatin accessibility landscape of S-SVF and V-SVF

To determine how the chromatin accessibility landscape changes between S-SVF and V-SVF, we performed differential analysis of DHS signal intensity between S-SVF and V-SVF. We identified 1261 significantly different DHSs (P < 0.01) (Fig. 2A and Table S3-1), suggesting distinct chromatin accessibility landscapes between S-SVF and V-SVF. Differential DHSs are mainly located at gene bodies (42.2%) and proximal promoter regions (34.2%) (Fig. 2B). To determine the specific genes that were associated with the DHSs, we assigned DHSs to their closest genes if they were located in a gene body or promoter (−20 kb to +100 bp from TSS) region. One thousand sixty-one differential DHSs were assigned to 790 genes (Table S3-2). By using GO enrichment analysis, we found that genes with differential DHSs were enriched in functional categories related to neuronal development, metabolism of lipids and carbohydrates, immune processes, and inflammation (Fig. 2C). These biological processes are closely related to known differences in metabolic effects, and inflammatory potentials between SAT and VAT (Tchkonia et al., 2013; Tran & Kahn, 2010). For example, preadipocytes in SAT tend to have greater capacity for lipid accumulation than VAT (Tchkonia et al., 2006), and our results determined that some ELOVL fatty acid elongase family genes, such as ELOVL4 and ELOVL6, which are crucial to regulating lipid biosynthesis and insulin sensitivity (Matsuzaka et al., 2007; Shimano, 2012), had differential DHSs at their gene loci. Specifically, ELOVL4 exhibited a higher DHS signal intensity at gene promoter in S-SVF than that in V-SVF (Fig. 2D). This differential DHSs in the ELOVL family genes implies differences in gene transcriptional activity, which in turn, may cause different expression levels of ELOVL family genes, and may directly influence the capacity of lipid accumulation between S-SVF and V-SVF. The promoter of phosphatidylinositol-4,5-bisphosphate 3-kinase catalytic subunit beta (PIK3Cβ), which encodes an isoform of the catalytic subunit of phosphoinositide 3-kinase (PI3K), also shows strong DHS in S-SVF but not in V-SVF (Fig. 2D), indicating that the PIK3Cβ gene may have a stronger transcription activity in S-SVF than V-SVF. These data suggest that there is a reinforced signal transduction of the insulin-induced PI3K pathway in S-SVF, which is consistent with previous studies that found that SAT is associated with improved insulin sensitivity, while VAT is associated with insulin resistance (Lafontan & Berlan, 2003; Zierath et al., 1998). Moreover, the intron 1 of an adaptor protein, phosphotyrosine interacting with PH domain and leucine zipper 2 (APPL2), which is involved in impairing insulin sensitivity and inflammation (Cheng et al., 2014; Yeo et al., 2016), exhibited a strong DHS in V-SVF but not in S-SVF (Fig. 2D). This difference in DHSs in APPL2 may result in differential expression of APPL2, thereby influencing the regulation of APPL2, and subsequently, influence changes in insulin sensitivity and inflammation in S-SVF compared with V-SVF.

Figure 2 Differential chromatin accessibility landscape between S-SVF and V-SVF.

(A) Scatter plot showing the differential DHSs between S-SVF and V-SVF. (B) The genomic distribution of the differential DHSs. (C) Gene Ontology (GO) analysis of differential DHS-associated genes. (D) Genome browser view of ELOVLE4 PIK3CK, and APPL2 loci as representative examples of differential DHS-associated genes involved in regulating lipid metabolism and insulin sensitivity in WAT. Gray boxes indicate DHS sites. (E) KLF binding motifs were enriched in differential DHSs.

As DHSs are always bound by transcription factors (TFs), the differential DHSs could facilitate identification of some key TFs that may potentially regulate the differential DHS-associated genes. Therefore, we performed motif enrichment analysis in these differential DHSs. The results found motifs for the Krüppel-like factor (KLF) family of TFs, which have been reported to play a crucial role in regulating adipocyte metabolism and differentiation (Brey et al., 2009; Hsieh et al., 2019), were significantly enriched (Fig. 2E), suggesting different developmental traits of preadipocytes between SAT and VAT.

Linking accessible DHSs to H3K27Ac histone modification

Open chromatin structure, as indicated in our study by DHSs, are often associated with histone modifications (Heintzman et al., 2007; Wang et al., 2008). The histone modification H3K27Ac is a mark of transcription activation in mammals (Calo & Wysocka , 2013). To investigate whether our DNase-seq DHSs colocalized with the H3K27Ac histone modification, we performed H3K27Ac ChIP-seq using porcine SAT and VAT with two biological replicates, peak calling each ChIP-seq data. Reproducible peaks for the two biological replicates were identified by using the IDR, with 29,950 and 36,031 reproducible peaks identified in SAT (Table S4-1) and VAT (Table S4-2), respectively (IDR <0.01). Next, we examined the relative ChIP-seq signal intensity within 2 kb of the center of DHSs in SAT and VAT. The results found that H3K27ac signals were enriched around the DNaseq-seq DHSs (Fig. 3A). Moreover, we intersected the DNase-seq DHSs with H3k27a peaks and found that 65.4% and 27.3% of all DHSs overlap with H3k27a peaks (overlap >10 bp) in SAT and VAT, respectively (Fig. 3B and Table S5). Notably, the DHSs in gene proximal promoter frequently overlapped with H3k27a peaks in both SAT and VAT (94.9% and 85.5%), indicating that the DHSs in proximal promoters are more likely regulating transcription activation. However, the DHSs sites in intergenic regions rarely overlapped with H3k27a peaks (Fig. 3B). Together, these data suggest that there is a strong correlation between DNaseq-seq DHSs and H3K27ac sites, especially the DHSs in gene proximal promoters, and these data suggest a significant association between the transcriptional activation and H3K27ac histone modification.

Figure 3 Integrative analysis of DNase-seq and H3K27Ac ChIP-seq data.

(A) Heat map showing normalized H3K27Ac ChIP-seq reads present at center of DHSs in S-SVF (3,930 sites) and V-SVF(2801 sites). (B) Percent of all DHSs, promoter DHSs, gene body DHSs and intergenic DHSs overlapping H3K27ac peaks identified in SAT and VAT. (C) Scatter plot showing the differential H3K27ac peaks between S-SVF and V-SVF. (D) The Venn diagram showing the overlap of differential DHS-associated genes and differential H3K27ac ChIP-seq peaks-associated genes. (E) GO analysis of 104 genes in (D) using Metascape. (F) Genome browser showing DNase-seq signal and H3K27ac ChIP-seq signal around the PDPR loci. Gray boxes indicate DHSs and ChIP-seq peaks.

To determine whether SAT and VAT have differential H3K27Ac histone modification patterns, we performed differential analysis of H3K27Ac signal intensity between SAT and VAT. Three thousand one hundred sixty-two differential peaks were identified between SAT and VAT (P < 0.01) (Fig. 3C and Table S4-3). To gain insight into the genes regulated by differential peaks of H3K27ac, we assigned these peaks to annotated genes using the same criterion as those used in DNase-Seq. One thousand three hundred fifty-nine peak-associated genes were identified (Table S4-4). We intersected this gene dataset with the differential DHS-associated gene dataset to identify a set of 104 genes that contain both differential DHSs and differential H3K27ac peaks (Fig. 3D and Table S4-5). GO enrichment analysis on these 104 genes were mainly enriched in categories related to energy metabolism of adipocytes, immune processes, inflammation, and neuronal development (Fig. 3E). Notably, an accessible region in the TSS of pyruvate dehydrogenase phosphatase regulatory subunit (PDPR), which encodes the regulatory subunit of pyruvate dehydrogenase phosphatase (PDP), had stronger signal intensity in S-SVF. Consistently, a higher H3K27ac peak was observed at same site in SAT (Fig. 3F), suggesting that the DHS was modified by H3K27ac, and this region may act as an enhancer to promote PDPR gene transcription in S-SVF. PDP is a crucial activator for pyruvate dehydrogenase complex (PDC),which plays a central role in linking glycolysis pathway to the oxidation process of tricarboxylic acid cycle and fatty acid synthesis (Patel & Korotchkina, 2006). Thus, the different transcriptional regulation of the PDPR gene may cause differential activation of PDC, further influencing the differences in the energy metabolism processes between SAT and VAT.

Linking differentially expressed genes to differential DHSs

Gene expression regulation is often influenced by multiple cis-regulatory elements. To investigate the relationship between differential DHSs identified by DNase-seq and differentially expressed genes, we performed RNA-seq on porcine SAT and VAT with two biological replicates. The detailed information and sequencing depth of RNA-seq are listed in Table S6. Differential analysis of global gene expression profiles between SAT and VAT was performed, and 1,784 differentially expressed genes (DEGs) were identified (P < 0.01), including 736 up-regulated genes in SAT and 1,048 up-regulated genes in VAT (Fig. 4A and Table S7). The results of a power analysis calculation shown that the estimated power is 8.12 with two biological repeats in RNA-seq. We compared the DHS signal intensity distribution in DEGs and non-DEGs within 5 kb upstream and downstream in both S-SVF and V-SVF. We found that the DHS signal intensity of DEGs were higher than non-DEGs throughout promoter, gene body, and downstream regions of gene transcription end site (TES) in both S-SVF and V-SVF, especially around the TSS. Specifically, the DHS signals of DEGs is 24.9% and 20.7% higher than that of non-DEGs in S-SVF and V-SVF, respectively (Fig. 4B), indicating that the chromatin around TSS of DEGs has a higher accessibility and contains more cis-regulatory elements than non-DEGs. Thus, we concluded that differential expression in the DEGs was more likely to be explained by the different chromatin accessibility in these DEGs between SAT and VAT.

Figure 4 Integrative analysis of DNase-seq and RNA-seq data.

(A) Volcano plot showing differentially expressed genes (DEGs) between SAT and VAT. (B) The distribution of DHS signals around DEGs (red) and non-DEGs (blue) in S-SVF (right panel) and V-SVF (left panel). (C) The Venn diagram showing the overlap of differential DHS-associated genes, differential H3K27ac ChIP-seq peaks-associated genes, and DEGs. (D) GO analysis of 20 genes in (C) using Metascape. (E) Genome browser showing DNase-seq signals, H3K27ac ChIP-seq signals, and RNA-seq expression profiles around the NR1D1 and CRYM loci. Gray boxes indicate DHSs and ChIP-seq peaks.

To further explore key functional genes that could cause differential biological characteristics between SAT and VAT, we intersected the DEG dataset with differential DHS-associated gene dataset and differential H3K27ac peaks-associated gene dataset to identify a set of 20 genes, which showed corresponding differential DHSs, differential H3K27ac peaks, and differential gene expression (Fig. 4C, Fig. S1 and Table S8). GO enrichment analysis of these 20 genes was enriched in immune processes and lipid metabolism (Fig. 4D). Notably, two energy metabolism associated genes were identified, nuclear receptor subfamily 1 group D member 1 (NR1D1) and Crystallin Mu (CRYM) (Fig. 4C, highlighted in red). NR1D1 is a key regulatory gene of adipogenesis and lipid metabolism (Marciano et al., 2014). The DNase-seq data found two DHSs, one at the proximal promoter and one in intron 1, in S-SVF that were not present in V-SVF. Moreover, the H3K27ac ChIP-seq data showed there were strong H3K27ac peaks across the proximal promoter, TSS, and part of intron 1 of NR1D1 in SAT but not in VAT. Simultaneously, the RNA-seq data found higher expression levels of NR1D1 in SAT than VAT (Fig. 4E). CRYM has been reported to regulate insulin sensitivity in human WAT (Serrano et al., 2014). The proximal promoter of CRYM showed a strong DHS in S-SVF but not in V-SVF, at the same location of genome. Additionally, SAT had a strong H3K27ac signal but VAT did not, while RNA-seq data showed that CRYM was highly expressed in SAT, but was almost not expressed in VAT (Fig. 4E). The higher expression of NR1D1 and CRYM in SAT could be explained by higher chromatin accessibility and H3K27ac signal around gene TSS in SAT. Taken together, chromatin accessibility could be a key regulator of gene expression in SAT and VAT, and these differences could lead to the distinct lipid metabolism and immune processes present in these two adipose tissue types.

Discussion

The SVF from WAT have depot-specific adipogenic potential, gene expression, and inflammatory cytokines secretion profiles, and understanding the regulation of the these factors are important to establish the pathophysiological heterogeneity between SAT and VAT (Silva & Baptista, 2019). Understanding the chromatin regulatory network is key to understanding the regulation of tissue-specific gene expression. In this study, we established and compared the chromatin accessibility profile for porcine S-SVF and V-SVF by using liDNase-seq to reveal the differences in the chromatin regulatory landscapes between S-SVF and V-SVF. We identified 3930 and 2801 DHSs in S-SVF and V-SVF, respectively (Table S1). The DHSs in S-SVF were mainly located in proximal promoters (50.9%), but DHSs in V-SVF were mainly located in gene bodies (42.1%, Fig. 1B), indicating the different chromatin accessibility patterns between S-SVF and V-SVF. The DHSs differential analysis identified 1261 DHSs that exhibited differential chromatin accessibility (Fig. 2A and Table S3-1). These differences indicate a global difference on transcriptional regulation between S-SVF and V-SVF. Notably, the differential DHSs were mainly distributed in proximal promoters (34.18%) and gene bodies (42.19%) (Fig. 2B), suggesting that proximal transcriptional regions played a significant role in forming differential gene regulation patterns between S-SVF and V-SVF.

DHSs are most significantly associated with the gene expression in various cells and tissues (Frank et al., 2015; Gonzalez, Setty & Leslie, 2015; Lu et al., 2016; Sieber et al., 2019). By integrating DNase-seq data with RNA-seq data, we found that the chromatin near the DEGs showed higher accessibility than that near the non-DEGs, especially at gene TSSs (Fig. 4B), suggesting the transcriptional regulation of the DEGs were more dynamic than non-DEGs. It could explain why DEGs are differentially expressed between SAT and VAT. Together, our results demonstrated that the chromatin accessibility landscapes had a profound effect on gene expression and play an important role in regulating gene expression in SAT and VAT.

SAT and VAT are functionally heterogeneous and contribute differently to metabolic diseases depending on their differential capacities of lipolysis, lipogenesis, insulin sensitivity, and secretion of inflammatory cytokines (Lafontan & Berlan, 2003). One recent study found that SAT and VAT had different sympathetic innervation, which would play a key role in regulating differential abilities of lipolytic and thermogenesis between SAT and VAT (Chi et al., 2018). Given the similar metabolic characteristics to humans, using pigs as the biomedical model to study human energy metabolism and obesity is therefore of significant value. Although previous studies using pig as the model have shown that SAT and VAT had differential gene expression profiles (Zhou et al., 2013), microRNA transcriptomes (Ma et al., 2013), and DNA methylomes profiling (Li et al., 2012), the effects of chromatin structure and the specific associations of function had not been investigated. In our results, we found that the differential DHSs-associated genes were enriched in the functional categories related to neuronal development, metabolism of lipid and carbohydrate, immune processes, and inflammation (Fig. 2C), which is consistent with the different features noted between SAT and VAT, in that SAT is associated with energy metabolism, whereas the VAT is associated with impaired inflammatory processes (Zhou et al., 2013). It is shown that liDNase-seq could provide adequate information to reflect specific features of fat depot type. Moreover, through motif enrichment analysis using genomic sequence of differential DHSs, we identified the KLF family of TFs preferentially bound differential DHS sites (Fig. 2E). This is consistent with the known role of various KLF family members in the regulation of adipocyte differentiation and adipogenesis in WAT. For example, KLF3 can inhibit preadipocyte differentiation by directly repressing the C/EBPα promoter (Bell-Anderson et al., 2013; Sue et al., 2008), and KLF4 is an critical early regulator in adipogenesis by inducing C/EBPβ (Birsoy, Chen & Friedman, 2008). Additionally, KLF6 positively regulates adipocyte differentiation by repressing delta-like 1, which is a negative regulator of adipocyte differentiation (Li et al., 2005). Thus, KLFs might be the key TFs involved in causing different capacity for adipogenesis in preadipocytes between SAT and VAT (Tchkonia et al., 2013). Together, these observations suggest that the chromatin accessibility landscapes play an important role in forming different metabolic and inflammatory features between SAT and VAT.

Our study integrated the data of chromatin accessibility, histone modifications, and transcriptomes to investigate the role of the chromatin regulatory landscape in controlling tissue-specific features found in SAT and VAT. We found that the DHSs strongly correlated with H3K27ac peaks. Some DHSs overlapped with H3K27ac peaks, indicating that these DHSs contained H3K27ac modifications, and were enriched at enhancer elements to promote gene transcription, especially in proximal promoter regions (Fig. 3B). Notably, the DHSs in SAT more frequently overlapped with H3K27ac peaks than that in VAT (Fig. 3B), revealing that the DHSs in SAT were occupied mainly by positive transcriptional elements, which indicates a higher transcriptional activity in SAT. By combining data from chromatin accessibility, histone modifications and transcriptomes assays, we identified a small group of genes that might be relevant to the phenotype characteristics of SAT and VAT (Fig. 4C). They harbored differential DHSs and H3K27ac peaks at their proximal transcription regulatory regions, and were accompanied by differential gene expression between SAT and VAT. Notably, the variation tendency of gene expression was consistent with the trend of DHS and H3K27ac signal changes in each fat depot (Fig. 4E). This established a classic model of gene transcriptional regulation, where chromatin is opened to become more accessible, the histones are modified with the H3K27ac modification and bound with TFs, which together facilitate the upregulation of gene transcription. NR1D1, which encodes REV-ERBα, was one of the genes identified in this study, and is expressed in adipose tissues and has been demonstrated to play a crucial role in lipid metabolism and glucose homeostasis (Kojetin & Burris, 2014; Marciano et al., 2014). REV-ERBα-deficient mice showed metabolic disorders with dyslipidemia (Raspe et al., 2002) and overall adiposity (Delezie et al., 2012). A second gene identified in this study, CRYM, has an essential role in mediating glycolipid metabolism and insulin sensitivity in WAT, and is expressed at a higher level in SAT than VAT (Ohkubo et al., 2019; Serrano et al., 2014). In our study, the higher expression of NR1D1 and CRYM in SAT, which is regulated by chromatin accessibility and H3K27ac modification, could cause higher lipogenesis potential and insulin sensitivity in SAT than VAT. Thus, these two genes could be the key functional genes that regulate the distinct metabolic characteristics in SAT and VAT. Our study established the chromatin regulatory landscapes and disclosed how they interact with epigenetic signature to achieve profound change of gene expression. It is important for understanding the functional differences between SAT and VAT and their different effects in obesity and metabolic diseases. However, our study bears some limitations. First, two biological replicates were used in our study, it would limit our ability to accurately identify differential genes with small fold change. A second limitation is that candidate genes found in our study were only based on high-throughput omics assays. The lack of in vivo/vitro functional validations are necessary for revealing the regulatory mechanism of depot-specific characteristics in SAT or VAT. It is likely that these limitations will be overcomed with the ongoing accumulation of additional epigenetics data and the further research at cellular level to verify the concrete functions of these genes.

Conclusions

The differences in chromatin regulatory landscapes between S-SVF and V-SVF are critical factors in regulating differential gene expression and forming different metabolic and inflammatory features between SAT and VAT. Furthermore, we found 20 key functional genes had different chromatin accessibility, H3K27ac modification and expression level between SAT and VAT. Meanwhile, some of these genes (such as NR1D1 and CRYM) are associated with the metabolic or inflammatory difference between SAT and VAT. Thus, these genes could be the crucial candidate genes to regulate depot-specific characteristics in SAT and VAT.

Supplemental Information

Supplemental Information 1 The genomic locations of DHSs in S-SVF and V-SVF

Click here for additional data file.

Supplemental Information 2 The genomic distribution of the DHSs

Click here for additional data file.

Supplemental Information 3 The genomic location and annotation of differential DHSs

Click here for additional data file.

Supplemental Information 4 Analysis of the H3K27Ac ChIP-seq data

Click here for additional data file.

Supplemental Information 5 The overlappde H3K27ac peaks in SAT and VAT

Click here for additional data file.

Supplemental Information 6 Statistics and mapping results of RNA-seq data

Click here for additional data file.

Supplemental Information 7 Up-regulated genes in SAT and VAT

Click here for additional data file.

Supplemental Information 8 The functions of 20 genes in Fig. 4C

Click here for additional data file.

Supplemental Information 9 Genome views of the DNase-seq, ChIP-seq and RNA-seq data for the 18 genes in Fig. 4C

Genome browser showing DNase-seq signals, H3K27ac ChIP-seq signals, and RNA-seq expression profiles around the 18 genes in Fig. 4C. Gray boxes indicate DHSs and ChIP-seq peaks.

Click here for additional data file.

Supplemental Information 10 Author Checklist

Click here for additional data file.

Additional Information and Declarations

Competing Interests

Author Contributions

Animal Ethics

Data Availability

The authors declare there are no competing interests.

Wenyang Sun conceived and designed the experiments, performed the experiments, analyzed the data, prepared figures and/or tables, authored or reviewed drafts of the paper, and approved the final draft.

Tinghuan Zhang performed the experiments, analyzed the data, prepared figures and/or tables, and approved the final draft.

Silu Hu analyzed the data, prepared figures and/or tables, and approved the final draft.

Qianzi Tang analyzed the data, authored or reviewed drafts of the paper, and approved the final draft.

Xi Long and Xu Yang performed the experiments, prepared figures and/or tables, and approved the final draft.

Shuangbao Gun conceived and designed the experiments, authored or reviewed drafts of the paper, and approved the final draft.

Lei Chen conceived and designed the experiments, analyzed the data, prepared figures and/or tables, authored or reviewed drafts of the paper, and approved the final draft.

The following information was supplied relating to ethical approvals (i.e., approving body and any reference numbers):

Animal Ethical and Welfare Committee of Gansu Agricultural University provided full approval for this research (approval number: AEWC-GAU-2019-096).

The following information was supplied regarding data availability:

The data is publicly available at NCBI SRA: PRJNA776562.

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
