# Peer review of "Chromatin accessibility landscape of stromal subpopulations reveals distinct metabolic and inflammatory features of porcine subcutaneous and visceral adipose tissue"

_PeerJ, doi:10.7717/peerj.13250_

## Round 0.1 · original submission · Major Revisions

Please address concerns of both reviewers and revise your manuscript accordingly.

Reviewer 1 ·

Basic reporting

1) The manuscript is written in clear and unambiguous, professional English and requires only minor corrections.
2) Literature references are sufficient to provide the scientific context
3) The figures and tables do not raise any concerns.
Regarding the article structure:
- The Abstract would benefit from the "Conclusion(s)" section.
- In the Discussion, at least some limitations of the study should be mentioned
4) The study design should be improved to increase the value of the results.

Experimental design

In their work, Wenyang Sun et al. studied the impact of differences in chromatin accessibility on gene expression in adipose tissue-derived stromal vascular fractions (SVFs) from SAT and VAT in a porcine model of diet-induced obesity. They found that distinct chromatin accessibility patterns may play a significant role in transcriptional regulation in these tissues. The study concept is interesting, and all molecular and bioinformatic analyses do not raise any concerns. However, some points regarding the study design impact the value of the obtained findings.
1) The tissues analyzed in the study were obtained from just two animals. Was the number of studied animals sufficient to get reliable results?
2) To verify if the study findings are obesity-specific, control experiments on tissues obtained from non-obese animals on a standard diet should be performed.

Validity of the findings

The above-mentioned methodological limitations decrease the value of the study findings.

Additional comments

Minor revisions:
Abstract:
1) "Subcutaneous adipose tissue (SAT) and visceral adipose tissue (VAT) have distinct
characteristics in adipose metabolism and impaired inflammation response, thus causing their different effects in metabolic disorders.” – Impaired inflammatory response refers only to dysfunctional adipose tissue, e.g., in the course of obesity, not to SAT and VAT of healthy subjects

2) The Abstract would benefit from the "Conclusion(s)" section.

Introduction:

1) “Generally, two major white adipose tissues (WATs) are recognized: subcutaneous adipose tissue (SAT) and visceral adipose tissue (VAT)” – in the literature, SAT and VAT are considered as distinct WAT depots, not different types of WAT.
2) When describing the functional differences between VAT and SAT, the Authors should keep a balance and at least mention the works that point at the contribution of SAT dysfunction to the development of metabolic complications of obesity.
3) This section of the manuscript should contain a formulated aim of the study.


Whole manuscript:
- Please explain abbreviations as they occur in the text for the first time, e.g., DHS-associated, IL, etc.

Annotated reviews are not available for download in order to protect the identity of reviewers who chose to remain anonymous.

Reviewer 2 ·

Basic reporting

The article is easy to follow. I have a few minor comments on the writing:
1. In the summary of the results in the beginning of the manuscript (line 37), the authors use abbreviations SVFs, SAT and VAT. I would prefer the authors write them out here; for a reader who is just looking over the first page summary it is easier to follow with the abbreviations written out in the main text. The abbreviations are fine in the other parts of the manuscript.
2. The authors in some cases write out numbers in words (line: 290,293,357) and in some cases use just the numeric numbers (line: 324, 325 ). I ask the authors to be consistent in one format.
3. The numbering of the supplemental table and the text are not matching up:
Line 226 should refer to table 3 not 2
Line 275, 276 should refer to table 4
Line 296, 294 also has the wrong table numbering
Please make sure the numbering is consistent (so that the right data is referenced in the text).
4. The supplemental table lacks some more annotation so it is easier to follow what is presented in each column.

The figures in the manuscript are overall easy to follow and understand. I have the following minor suggestions:
1. For the heat maps in figure 1A and figure 3A the legend number should be in a larger font and the z-axis is missing the label.
2. In Figure 2C and Figure 3E the label of the y-axis are in a faint gray color and hard to read. I would suggest increasing the font size and making the text black.

The relevant raw data is available in the supplemental data.

Minor comments:
In the introduction (line 48), a reference to the threat posed by obesity to global health is missing. Overall the introduction could include some of the more recent publications in the field e.g.

Karlsson T, Rask-Andersen M, Pan G, et al. Contribution of genetics to visceral adiposity and its relation to cardiovascular and metabolic disease. Nat Med 2019; 25: 1390–95. 65
Marinou K, Hodson L, Vasan SK, et al. Structural and functional properties of deep abdominal subcutaneous adipose tissue explain its association with insulin resistance and cardiovascular risk in men. Diabetes Care 2014; 37: 821–29

Otherwise the introduction clearly sets up the research question.

Experimental design

The methods used in this study are clearly described in the methods section and the study is conducted with satisfactory technical and ethical standards.
The authors use two Bama miniature pigs to investigate the difference in chromatin accessibility between subcutaneous adipose tissue (SAT) and visceral adipose tissue (VAT). The authors use a previously established modified DNase-seq protocol called liDNase-seq that allows de novo mapping of DHS with only a small number of input cells.
It has been known that VAT is biochemically different from SAT and the expression of inflammatory associated features in VAT are speculated to be responsible for its association with diseases. The authors’ findings tie into these hypotheses by reporting on differences in cis-regulatory elements between SAT and VAT.

Comments on the results/data of the paper:

1. The samples used are limited and therefore the data has only 2 biological replicates. I would like the authors to discuss this limitation briefly in the discussion section, including any variability between the replicates.
2. The authors comment on genes that are identified in the DNase-seq analysis of chromatin accessibility such as ELOVLE4 and the promoter region of PIK3Cβ, but do not mention whether these genes are identified in the RNA-seq experiment. If the DNase-seq experiment suggests up-regulation of the genes, evidence should be seen in the RNA-seq data. The authors should comment on that. The same is true for the finding of PDPR from the H3K27C-ChiP-seq experiment.
3. Furthermore, the GO term analysis of the data finds up-regulation of genes involved in neuron development. This seems to be an unexpected finding - I would like the authors to comment on that finding and discuss it in the context of the literature.
4. It is interesting that only a small number of 20 genes are found to overlap in all 3 analyses. The authors only focus on 2 genes (CRYM, NR1D1) that are explored and discussed in more detail, which is suitable for the main text of the manuscript given the genes’ known functions. Nevertheless, for the interested reader it would be great to provide some context of the known functions of the other 18 genes as well as presentation of the data such as shown in Figure 4E for CRYM and NR1D1. These can be attached in a supplemental table and figure.

Validity of the findings

In the context of the two biological replicates, the findings and data analysis of the manuscript are good and valid. The data tie into the reported findings in the literature and add new investigation of cis-regulatory elements in adipose tissue. More biological replicates would lead to a more robust analysis, but that may be outside of the scope of this manuscript. The selection of pigs as a model system makes this study relevant to human physiology.

---

## Round 0.2 · accepted · Accept

All critiques of the reviewers were adequately addressed and the manuscript was revised accordingly. Therefore amended version is acceptable now.

Reviewer 1 ·

Basic reporting

no comment

Experimental design

no comment

Validity of the findings

no comment

Additional comments

I want to express my gratitude for the opportunity to re-review the paper entitled: " Chromatin accessibility landscape of stromal subpopulations reveals distinct metabolic and inflammatory features of porcine subcutaneous and visceral adipose tissue" by Wenyang Sun et al. Since the authors addressed my concerns regarding the methodology as well as some minor revisions, I find the manuscript acceptable for publication.

Reviewer 2 ·

Basic reporting

The revised manuscript has addressed my previous concerns.

Experimental design

The revised manuscript has addressed my previous concerns.

Validity of the findings

The revised manuscript has addressed my previous concerns.

Additional comments

The authors have addressed my comments in their written response and changed the manuscript to reflect that. The authors now address the limitation of the study in an honest way, which will help the reader to put it into context. The authors have fixed the numbering and information in the supplementary data.
I think the manuscript can be published with the included changes.